# Interferon Tau (IFNt) and Interferon-Stimulated Genes (ISGs) Expression in Peripheral Blood Leukocytes and Correlation with Circulating Pregnancy-Associated Glycoproteins (PAGs) during Peri-Implantation and Early Pregnancy in Buffalo Cows

**DOI:** 10.3390/ani12223068

**Published:** 2022-11-08

**Authors:** Anna Beatrice Casano, Vittoria Lucia Barile, Laura Menchetti, Gabriella Guelfi, Gabriele Brecchia, Stella Agradi, Giovanna De Matteis, Maria Carmela Scatà, Francesco Grandoni, Olimpia Barbato

**Affiliations:** 1Department of Veterinary Medicine, University of Perugia, Via San Costanzo, 06126 Perugia, Italy; 2Research Centre for Animal Production and Aquaculture, Consiglio per la Ricerca in Agricoltura e l’Analisi dell’Economia Agraria (CREA), Via Salaria 31, 00015 Monterotondo (Roma), Italy; 3School of Bioscience and Veterinary Medicine, University of Camerino, Via Fidanza, 62024 Matelica (Macerata), Italy; 4Department of Veterinary Medicine, University of Milano, Via dell’Università, 26900 Lodi, Italy

**Keywords:** interferon-stimulated genes (*ISGs*), PBMCs, PMNs, *IFNt*, PAGs, pregnancy, buffalo

## Abstract

**Simple Summary:**

The peri-implantation period is a particularly delicate moment of pregnancy. To better elucidate the dialogue between the conceptus and uterine endometrium and identify a potential strategy to improve embryo survival, we have analyzed the interferon-stimulated genes (*ISGs*) and interferon tau (*IFNt*) expression in peripheral blood mononuclear cells (PBMCs: lymphocytes and monocytes) and polymorphonuclear leukocytes (PMNs: granulocytes) during the peri-implantation period and until 40 days of pregnancy in buffalo cows. Additionally, we have evaluated the possible relationship between the expression of these genes and peripheral plasma concentration of pregnancy-associated glycoproteins (PAGs).

**Abstract:**

The objective of this study was to analyze interferon-stimulated genes (*ISGs*) and interferon tau (*IFNt*) gene expression in peripheral blood leukocytes during the peri-implantation period and until 40 days of pregnancy in buffalo cows. Relationships were also examined between the expression of *ISGs* and *IFNt* and pregnancy-associated glycoproteins (PAGs) peripheral plasma concentration. Buffalo cows were synchronized and artificially inseminated (d 0). Blood samples were collected on days 0, 18, 28 and 40 after artificial insemination (AI) for peripheral blood mononuclear cells (PBMCs) and polymorphonuclear leukocytes (PMNs) isolation and PAGs radioimmunoassay analysis. The study was carried out on 21 buffalo cows divided ex post into Pregnant (*n* = 12) and Non-pregnant (*n* = 9) groups. Steady state levels of *OAS1*, *MX2*, *ISG15* and *IFNt* mRNA were measured by RT-qPCR and their estimated marginal means (*p* < 0.01 for all) were higher in pregnant than non-pregnant buffaloes, both in PBMCs and PMNs. In PBMCs, pairwise comparisons showed that *OAS1* and *MX2* expressions differed between pregnant and non-pregnant buffaloes on all the days of observation (*p* < 0.001), while significant differences in *ISG15* and *IFNt* started from day 28 post-AI (*p* < 0.05). In PMNs, *ISG15* expression differed between groups only at days 18 and 28 (*p* < 0.001), while comparisons were always significant for *IFNt* (*p* < 0.05). The expression of all genes, except *ISG15* as determined in PMNs, was positively associated with PAGs plasma concentrations (*p* < 0.05). This work showed a significant increase in *ISGs* and *IFNt* expressions in PBMCs and PMNs in buffalo during the peri-implantation period and early pregnancy, and their correlation with PAGs plasma concentration.

## 1. Introduction

The early events occurring during the phases of blastocyst implantation represent a pivotal moment for the maintenance of pregnancy. During the peri-implantation period, the developing blastocyst depend on histotrophic secretion inside the uterus [1,2], comprised of a mixture of nutrients, enzymes, growth factors, hormones and transport proteins regulated by embryo-maternal cross-talk [3,4]. One important signal to the maternal system to sustain pregnancy recognition is interferon tau (*IFNt*). *IFNt* is one of the first molecules involved in the mechanism of early maternal recognition in ruminants [5]. This protein is secreted by the trophectodermal cells of the blastocysts around days 14–15 of pregnancy in cattle [6], days 13–17 in sheep [7], and days 16–25 in buffalo [8], and increases with the elongation of the conceptus [9,10,11]. This type of interferon is characterized by antiviral, antiproliferative and immunomodulatory properties, and therefore controls luteotropic and immune mechanisms for successful embryo implantation [12]. Furthermore, *IFNt* induces perceptible temporal changes in local and peripheral tissues during its release [11,13,14]. It acts in the uterus to prevent luteolysis by inhibiting prostaglandin F_2α_ release, resulting in the maintenance of corpus luteum function [15]. In fact, *IFNt* stimulates the expression of interferon-stimulated genes (*ISGs*), including *interferon-stimulated gene 15 ubiquitin-like modifier* (*ISG15*), *Myxovirus resistance 2* (*MX2*), and *2′,5′-oligoadenylate synthetase 1* (*OAS1*), in various cells, such as endometrial, luteal, and peripheral blood cells [16,17]. In bovine, several studies have indicated that the expression of ISGs increase during early gestation in peripheral blood leukocytes [18,19,20,21], whereas few data are available for buffalo. Thakur et al. [22] reported the expression of *ISGs* (*OAS1*, *MX1*, *MX2* and *ISG15*) in buffalo peripheral blood mononuclear cells (PBMCs) during the peri-implantation period finding that most ISGs increase through day 14 to 20 post-artificial insemination (AI).

In the ruminant species exist other molecules potentially involved in the maintenance of pregnancy, named pregnancy-associated glycoproteins (PAGs). These glycoproteins (sub-class of aspartic proteases) [23,24] are synthesized by the mono- and bi-nucleated trophoblast cells of the eutherian species, including buffalo [25,26,27,28], and released into the maternal blood at the time of implantation [29]. Roberts et al. [30] showed a possible role for PAGs in binding and sequestering peptides susceptible to recognition by the major histocompatibility complex (MHC) and exerting an immunomodulatory role at the maternal-fetal level, necessary for the formation and preservation of the maternal-fetal unit histocompatibility. Dosogne et al. [31] suggested that the trophoblast PAGs production could be a mechanism by which the conceptus protects itself from rejection. Other authors [32,33] supposed a luteotropic role of PAGs as in vitro trials showed that they induce the release of prostaglandin (PG) E_2_ and progesterone from luteal cells, and PGE2 from endometrial cells. Austin et al. [34] attributed to PAGs a hormonal role in inducing the release of granulocyte chemotactic protein-2 (GCP-2), an alpha chemokine whose synthesis is induced by *IFNt* in early pregnancy. The PAGs detection in maternal blood of ungulate ruminants has become a useful tool for monitoring pregnancy. Moreover, PAGs could be considered an indicator of the viability of the fetal-placental unit in ruminants, and therefore can be used for early detection of placental alterations and embryonic losses [29,35,36]. In buffalo cow, different RIA systems were utilized to detect PAGs in maternal blood [28,37,38,39]. In the last decades, a system involving the use of antisera against buffalo PAGs for the development of RIA systems and detection of pregnancy has been created [27,28]. In pregnant buffalo, the plasma concentrations of PAGs are detectable starting from 25 days after conception [26,39].

Understanding the molecular pathways involved in the survival of the embryo during the early stages of its life could be useful to increase reproductive efficiency in livestock species. This is even more true in buffalo cows, where a lower efficiency is showed during the daylight lengthening period, i.e., spring–summer period (low-breeding season) [35,40]. In this species, to ensure continuity in the calving and production throughout the year, the mating also occurs in the low-breeding season although with a lower fertility.

From our current knowledge, there are no studies on ISGs expression in both leukocytes cell types: PBMCs and PMNs, and their correlation with PAGs in buffalo. Thus, the objective of this study was to quantify interferon-stimulated genes (*OAS1*, *MX2*, *ISG15*) and *IFNt* expression in PBMCs and PMNs during peri-implantation and early pregnancy in buffalo cows. Additionally, the possible relationship between *ISGs* and *IFNt* mRNA expression and maternal blood concentration of PAGs was also investigated.

## 2. Materials and Methods

### 2.1. Animals and Experimental Design

The study was carried out at the CREA Animal Production and Aquaculture experimental farm in Monterotondo, Rome, Italy. The experimental procedures were assessed and approved by the CREA Committee of Ethics in Animal Research (Protocol N.0081676-02/11/2020).

A total of 21 animals belonging to the Italian Mediterranean buffalo herd subjected to a synchronization and AI program were enrolled in this study and grouped as described below. Regular clinical examination was performed, in particular before estrus synchronization to exclude diseases such as endometritis, mastitis and metabolic disorders.

The buffalo cows were synchronized with a progesterone-releasing intravaginal device (PRID^®^) associated to PMSG and PGF_2α_ analogues reported by Barile et al. [35] Animals were artificially inseminated using frozen-thawed semen at 72 h after PRID^®^ removal.

Blood samples were collected from the jugular vein in 10 mL EDTA tubes at days 0 (d 0), 18 (d 18), 28 (d 28) and 40 (d 40) from AI (AI = d 0) for the determination of PAGs plasma concentration and isolation of RNA for the gene expression analysis. For PAG determination, plasma was separated by centrifugation at 2700× *g* for 10 min and stored at 20 °C until assayed.

The animals were grouped *ex post* as Pregnant (*n* = 12) and Non-pregnant (*n* = 9), as ascertained by ultrasonography and PAGs plasma concentration at day 28 and 40 based on diagnostic criteria as described below in Section 2.2.

### 2.2. Pregnancy Diagnosis

Transrectal ultrasonography was done on d 28 and d 40 post-AI to diagnose pregnancy. Buffaloes were considered pregnant if an embryonic vesicle and embryo proper with beating heart were recognized, while in the absence of these signs buffaloes were considered as non-pregnant [35].

Based on the PAGs plasma value (cut-off value: 1 ng/mL), buffaloes were considered non-pregnant when concentrations remained very close to zero throughout sampling and pregnant when the concentrations were ≥1 ng/mL at d 28 and d 40 post-AI.

### 2.3. PAGs Radioimmunoassay

For the determination of PAGs plasma concentrations, the RIA system was applied as previously described by Barbato et al. [27] A bovine PAG67kDa preparation (boPAG67kDa, accession number A61232) was used as both standard and tracer [41]. The assay was performed in duplicate and the initial dilution for primary AS#860 was 1:300000. The intra- and inter-assay coefficients of variation were 2.7 and 7.9%, respectively.

### 2.4. Isolation of PBMCs and PMNs

The purification of PBMCs and PMNs was obtained following a density gradient using Lymphoprep™ (1.077 g/mL; Axis-Shield PoC AS, Oslo, Norway). The blood samples (10 mL) were taken from the external jugular vein and collected in vacutainer tubes containing anticoagulant EDTA. The protocol used for the isolation of cells was described by Barbato et al. [42]. In brief, 10 mL of whole blood was diluted 1:1 with Hank’s Buffered Salt Solution (HBSS) into a 50 mL Falcon^®^ tube and then carefully layered on 10 mL of Lymphoprep™. The tubes were centrifuged at 800× *g* for 30 min. The phase containing erythrocytes and PMN settled down at the bottom of the tube. The phase containing PBMC settled on the plasma: Lymphoprep™ interface. The upper plasma layer was removed and discarded without disturbing the plasma: Lymphoprep™ interface. The PBMC layer was collected at the plasma: Lymphoprep™ interface, without disturbing the erythrocyte/PMN pellet and washed twice with HBSS. Afterward, erythrocytes were lysed with 1:10 *v*/*v* of 1× ammonium chloride lysing solution and washed once with HBSS. The purity of the PBMCs and PMNs fractions were evaluated by flow cytometry based on the forward scatter channel (FSC) vs. the side scatter channel (SSC). The purities of PBMCs and PMNs were on average 98 and 85%, respectively.

The isolated PBMCs and PMNs were stored at −80 °C until RNA extraction.

### 2.5. Gene expression Level of IFNt and ISGs: RNA Isolation, Reverse Trascription and qPCR

Total RNA was isolated from PBMCs and PMNs using the Total RNA Purification kit (NorgenBiotek Corp™, Thorold, ON, Canada), following the manufacturer’s procedures. The samples were treated with RNase-free DNase I Kit (NorgenBiotek Corp™, Thorold, ON, Canada) to prevent genomic DNA contamination. The quality of RNA was assessed by A260/A280 ratio and quantified using the Nanodrop 2000 Spectrophotometer (Thermo Fisher Scientific Inc., Waltham, MA, USA), and the RNA was stored at −80 °C until analysis. A quantity of 100 ng of total RNA from each sample was reverse transcribed with iSCRIPT cDNA (Bio-Rad, Hercules, CA, USA), according to the manufacturer’s protocol. The cDNA obtained from each sample was used as a template for qPCR. The primers and probes were designed using the PrimerQuest™ Tool (Integrated DNA Technologies, Coralville, IA, USA) based on buffalo gene sequences taken from the NCBI database (Table 1). Relative quantification of *IFNt* and *ISGs* transcript was carried out following the MIQE guidelines [43].

Gene expression qPCR was performed as described by Filipescu et al. [44].

The relative expression genes were normalized to ACTB reference gene levels. The 2^−ΔΔCt^ method was used to calculate the relative expression of the target genes [45].

### 2.6. Statistical Analysis

Diagnostic graphics were used to check assumptions and outliers. PAGs concentration was Log(x + 1) transformed for the analyses [28]. Raw values were presented as means and standard errors. The data were analyzed using linear mixed models. Animals were included in the models as random while “Time” as repeated effects with a scaled identity covariance structure. The models evaluated the main effects of time (3 levels: 18, 28, and 40 days post-AI), AI outcome (2 levels: Pregnant and Non-pregnant), and their interaction. Sidak adjustment was used for carrying out multiple comparisons. Then, data were stratified according to the outcome and day of observation, and differences in gene expression between PBMCs and PMNs were investigated only on pregnant animals. Thus, these models only included the effect of the matrix (2 levels: PBMC and PMN). The Pearson coefficient (r) was used to evaluate the correlations between gene expression and PAGs concentration. The association was considered poor if r < |0.3|, medium if |0.3| ≤ r < |0.5|, or large if r ≥ |0.5| [46]. Statistical analyses were performed with SPSS 25.0 (SPSS Inc. Chicago, IL, USA), and statistical significance was set at α < 0.05.

## 3. Results

A total of 12 out of 21 buffalo cows enrolled in this study became pregnant while 9 remained non-pregnant as determined by PAGs plasma concentration and ultrasonography at days 28 and 40 post-AI.

### 3.1. PAGs Concentration in Pregnant and Non-Pregnant Groups

Differences in PAGs concentrations between Non-pregnant and Pregnant groups were found starting from d 28 of sampling, when they were 0.3 ± 0.1 ng/mL and 2.8 ± 0.8 ng/mL, respectively (*p* < 0.001). In non-pregnant buffaloes, PAGs concentrations remained constantly close to zero ng/mL throughout the sampling period, while in pregnant buffaloes it increased significantly from d 28 post-AI (*p* < 0.001).

### 3.2. ISGs Expression in PBMCs and PMNs in Pregnant and Non-Pregnant Groups

Estimated marginal means of *OAS1*, *MX2*, *ISG15* and *IFNt* were higher in pregnant than non-pregnant buffaloes (*p* < 0.01 for all), both in PBMCs and PMNs. In PBMCs, pairwise comparisons showed that *OAS1* and *MX2* expressions differed between Pregnant and Non-pregnant groups on all the days of observation (*p* < 0.001), while significant differences in *ISG15* and *IFNt* started from d 28 post-AI (*p* < 0.05; Figure 1). In PMNs, *ISG15* expression differed between groups only at days 18 and 28 (*p* < 0.001) while comparisons were always significant for *IFNt* (*p* < 0.05; Figure 2).

Significant effects of the Time (*p* < 0.001) and Time x Outcome interaction (*p* < 0.01) were found for ISG15, both in PBMCs and PMNs. Its expression in the Pregnant group peaked at day 28 post-AI when evaluated in PBMCs (*p* < 0.05, Figure 1), while it fell on d 40 when evaluated in PMNs (*p* < 0.001, Figure 2). In PMNs, changes over time were also found for *IFNt*. In particular, its expression increased at d 40 compared with the previous time point (*p* < 0.05). Expression levels of other genes (*OAS1* and *MX2*) remained constant over time (Figure 1 and Figure 2).

### 3.3. Differences between the PBMCs and PMNs Expression in Pregnant Group

Expression differences in the matrix were only found for ISG15, which was highest in PMNs at day 18 (*p* < 0.001) and PBMCs at day 40 (*p* = 0.001; Figure 3).

### 3.4. Correlations between Plasma Concentration of PAGs and Genes Expression

The expression of all genes, except *ISG15* as determined in PMNs, was positively associated with PAGs plasma concentration (*p* < 0.05; Table 2). The strongest correlations were found with *OAS1* in PBMCs (r = 0.705, *p* < 0.001) and *IFNt* in PMNs (r = 0.650, *p* < 0.001). *ISG15* was weakly associated with PAGs even when evaluated in PBMCs (r = 0.379, *p* = 0.039).

## 4. Discussion

To better understand the process involved in the maternal recognition and maintenance of pregnancy, the purpose of this work was to assess the gene expression of *ISG15, OAS1, MX2* and *IFNt* in maternal blood cells and their correlation with PAGs maternal plasma concentration during the peri-implantation period and early pregnancy in buffalo.

To our knowledge, no studies have been conducted to evaluate the mRNA expression of *ISGs* and *IFNt* in PBMCs and PMNs in buffalo cows in the early gestation period. Preliminary studies conducted in bovine showed that, in the early stages of pregnancy, the predominant class of genes that upregulated in leukocytes is represented by *ISGs* [19], in particular *ISG15*, *MX2* and *OAS1* [18,47].

In the present study, *OAS1* and *MX2* expression differed between the Pregnant and Non-pregnant group in all days of observation either in PBMCs or PMNs. These results agree with those previously reported in bovine in which was utilized either the total fraction of peripheral blood immune cells [13,18,48] or separate fraction of PBMCs and PMNs [46]. Different from our study in which data were recorded until d 40 post-AI, those observations ended 20 days post-AI.

In the buffalo species, Thakur et al. [22] and Mishra et al. [49] showed that the expression profile of *OAS1*, *MX2* and *ISG15* on PBMCs increased through days 14–20 post-AI and declined thereafter. In our study, we observed an increase in the expression of *OAS1* and *MX2* for all the time points of observation in PBMCs, while for *ISG15* the increase started from d 28 post-AI. The same profile was found in PMNs concerning the expression of *OAS1* and *MX2*, while the *ISG15* expression declined after d 28 post-AI and at d 40 no statistical differences between Pregnant and Non-pregnant groups were found anymore. According to different studies in ruminants [13,18,50], as well as in the present study, the *OAS1* and *MX2* exhibited the strongest expression during early pregnancy in the specific immune cell groups. Buragohain et al. [51] showed an increase of *MX2* gene expression in peripheral blood from days 14–28 until d 35 of pregnancy in buffalo. This increase is comparable to our finding on *MX2* gene expression both in PMNCs and PMNs. These results show a possible correlation between the expression of the investigated *ISGs* and progression of pregnancy in the buffalo.

Our results show an increase of *ISG15* expression started from d 28 post-AI in PBMCs. This is in contrast with the finding of Thakur et al. [22], which reported a significantly greater expression in pregnant buffaloes on days 18–24 post-AI. This outcome could be due to the difference in the methodology employed: when PBMCs are isolated, their degree of purity may vary due to possible PMNs contamination, whereby the RNA extraction will not only be from PBMCs but will contain contaminating PMN cells that could express the gene. We isolated the two leukocyte populations with 98% pure PBMCs. Regarding the *ISG15* expression in the PMNs, the trend found in our study is comparable with that reported by Thakur et al. in the PBMCs [22].

In bovine, different authors showed that circulating PMNs can respond earlier to *IFNt* stimuli [20,52,53]. Indeed, granulocytes and other cell types respond to *IFNt* by the same signaling pathway [54]; nevertheless, the reason why granulocytes appear to be more sensitive to *IFNt* is still unclear.

Regarding the *ISGs* expression at d 40 after AI, our results seem to be in contrast to those found in bovine by Sheikh et al. [55] and Panda et al. [56], which showed that the gene expression of *MX1*, *MX2*, *OAS1* and *ISG15* increase in blood between days 10–18 and then decline between days 20–36 of gestation. This could be due to the fact that in buffalo the *IFNt* is secreted by the trophectodermal cells of the blastocysts around days 16–25 [8], i.e., later with respect to cattle [6]. Some of the *IFNt* secreted escape the uterus and can be detected in blood [57,58]. For this reason, the *IFNt* in the blood is low [55], and therefore it is preferred to measure the response of circulatory leukocytes to *IFNt*, namely *ISGs* expression [18,48,59].

In fact, in our work, we found the expression of *IFNt* lower than that of *ISGs* in both PBMCs and PMNs, except for *ISG15* at d 40 post-AI. The expression of *IFNt* was significatively different between Pregnant and Non-pregnant groups starting from d 28 post-AI in PBMCs, while in PMNs the difference in the expression was evident since d 18 post-AI. This finding shows that the expression of *IFNt*, as that of *ISGs*, appears earlier in the PMNs cells.

Previous studies in bovine suggested an early response of PMNs to *IFNt* stimuli [16,20]. In vitro studies also suggested a greater sensitivity of PMNs to the stimulus of the conceptus because these cells respond quickly to low *IFNt* concentrations [53,54]. The results of our work do not support this hypothesis. Comparing the *ISGs* and *IFNt* expression between PBMCs and PMNs in pregnant buffaloes, we found a difference only for the *ISG15*, in agreement with Melo et al. [47] in bovine.

Concerning correlations between PAGs plasma concentration and genes expression, our results showed a strong correlation with *OAS1* and *IFNt* in both PBMCs and PMNs blood cells, differently from Dalmaso de Melo et al. [60] who found a low or a non-significant correlation between *OAS1* and *ISG15* with PAGs in bovine PMNs.

Our results support the possible immunomodulatory role of PAGs at the maternal-fetal level, essential for the formation and preservation of the maternal-fetal unit histocompatibility [31]. *IFNt* and PAGs could share a common role in preventing luteolysis by inhibiting PGF2α release, resulting in the maintenance of the CL function and consequently pregnancy [32,33,34]. Therefore, the late presence of gene expression that we found at d 40 post-AI in pregnant buffaloes could be linked to the fact that its function is not limited to the peri-implantation period, but could be implicated in the embryo survival mechanisms. Our preliminary study [42] showed the expression of mRNA *PAG-2* in peripheral leukocytes of pregnant buffaloes at days 14, 18, 28 and 40 of gestation, emphasizing the importance of the presence of PAGs at the time of the maternal recognition period. In fact, it has been reported that both *PAG-2* and *INFt* upregulated genes in bovine conceptus at day 14/day 21 of gestation [61].

## 5. Conclusions

The present study confirmed the *ISGs* expression during the peri-implantation period and early pregnancy, showing their possible connection not only in recognition and establishment of pregnancy but also in its maintenance in buffalo species.

The correlation between *ISGs* and *IFNt* expression and PAGs plasma concentration supports the possible immunomodulatory role for these glycoproteins at the maternal-fetal level, and the antiluteolytic function by inhibiting PGF_2α_ release, resulting in the maintenance of the CL function. Further research will be required to confirm these findings, as well as to verify a potential relationship between the quantity of *ISGs* and pregnancy failures.

## Figures and Tables

**Figure 1 animals-12-03068-f001:**
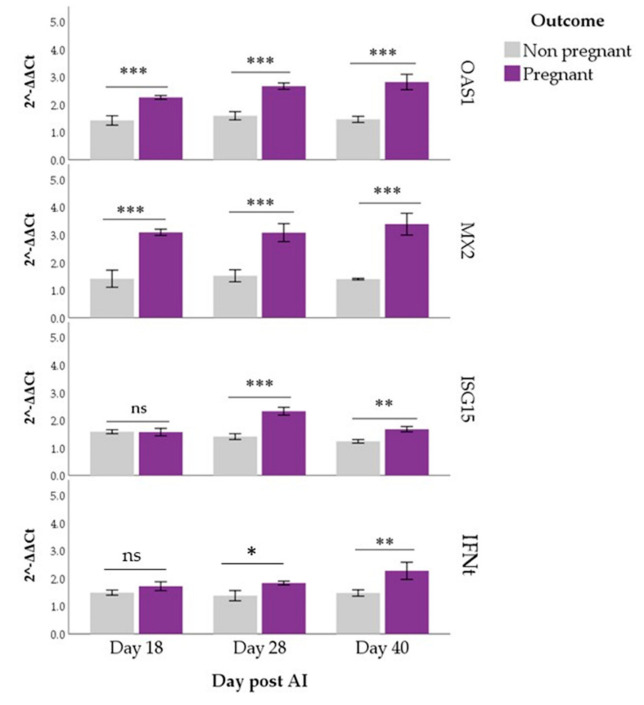
*OAS1*, *MX2*, *ISG15* and *IFNt* expression in PBMCs of pregnant and non–pregnant buffaloes. Asterisks indicate differences in expression between the Pregnant and Non-pregnant groups for each gene and each day post-artificial insemination (AI; * *p* < 0.05, ** *p* < 0.01, *** *p* < 0.001).

**Figure 2 animals-12-03068-f002:**
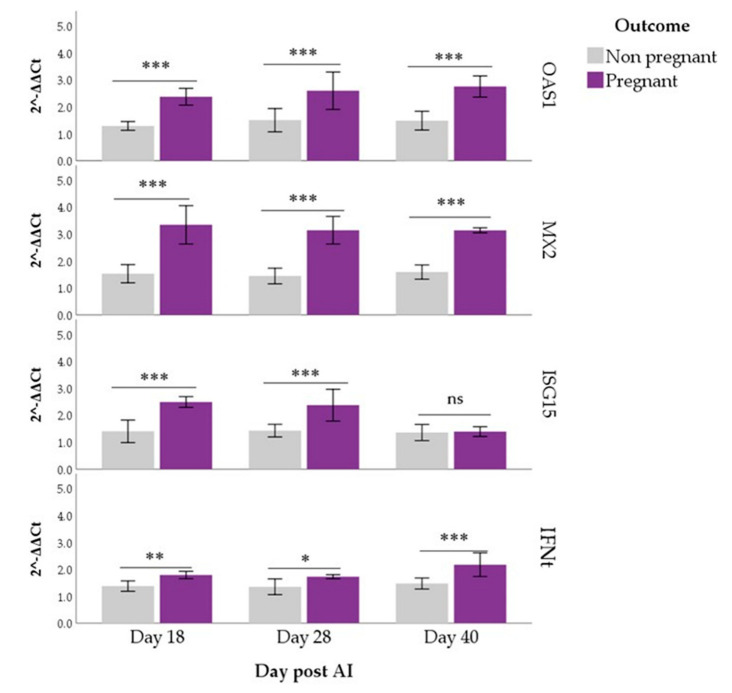
*OAS1*, *MX2*, *ISG15* and *IFNt* expression in PMNs of pregnant and non–pregnant buffaloes. Asterisks indicate differences in expression between the Pregnant and Non-pregnant groups for each gene and each day post-artificial insemination (AI; ns: not significant, * *p* < 0.05, ** *p* < 0.01, *** *p* < 0.001).

**Figure 3 animals-12-03068-f003:**
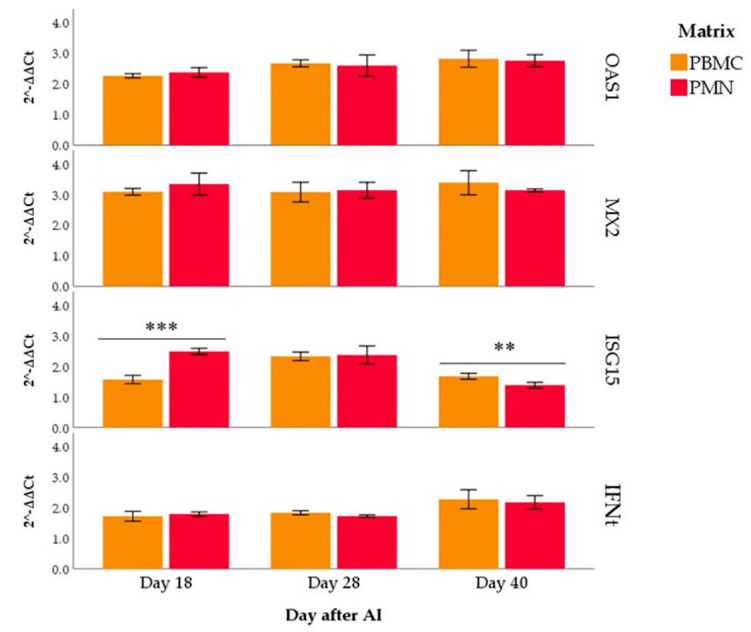
*OAS1*, *MX2*, *ISG15* and *IFNt* expression in polymorphonuclear leukocytes (PMNs) and mononuclear cells (PBMCs) of pregnant buffaloes. Asterisks indicate differences in expression between PBMCs and PMNs for each gene and each day post-artificial insemination (only significant differences are indicated; ** *p* < 0.01, *** *p* < 0.001).

**Table 1 animals-12-03068-t001:** The PrimeTime™ qPCR assays used in this study are listed. The PrimeTime™ qPCR assays are composed of a pair of unlabeled PCR primers and a probe with a 56-FAM dye label on the 5′ end.

Name	Sequence	NCBI RefSeq	Amplicon
** *IFNt* **		AY535404.1	99 bp
*Probe*	5′-/56-FAM/CCAGGGCAT/ZEN/CCATGTCTTCCTGAA/3IABkFQ/		
*Fw*	CCATTCTGACCGTGAAGAAGTA		
*Rev*	TCATCTCCACTCTGATGATTTCC		
** *ISG15* **		NM_001291322.1	95 bp
*Probe*	5′-/56-FAM/TGAGGGACT/ZEN/CCATGACAGTATCCGA/3IABkFQ/		
*Fw*	CTGAAGGTGAAGATGCTAGGG		
*Rev*	ATCTTCTGGGCGATGAACTG		
** *MX2* **		KM591576.1	100 bp
*Probe*	5′/56-FAM/AAGAGGCAC/ZEN/ACTCCGACTTTCCAC/3IABkFQ		
*Fw*	GTCATGTGGCTGTCCTTCA		
*Rev*	TGGCTGCTCATGGAAGTAAA		
** *OAS1* **		XM_025267539.1	88 bp
*Probe*	5′/56FAM/AGCGCCGAG/ZEN/GAGAATTCATCGAAG/3IABkFQ/		
*Fw*	GTCGTCTTCCTCACCAATCTC		
*Rev*	CTTCCAGCTGTCTCCTGATTT		
** *ACTB* **		NM_001290932.1	99 bp
*Probe*	5′/56-FAM/TGGCACCCA/ZEN/GCACAATGAAGATCA/3IABkFQ		
*Fw*	CGGACAGGATGCAGAAAGA		
*Rev*	TACTCTGTGTGGATTGGCG		

**Table 2 animals-12-03068-t002:** Pearson coefficients evaluating the associations between plasma concentration of PAGs and genes expression. *** *p* < 0.001, ** *p* < 0.01, * *p* < 0.05.

Gene	PMNs	PBMCs
**OAS1**	0.705 ***	0.611 ***
**MX2**	0.639 ***	0.566 **
**ISG15**	0.379 *	0.055
**IFN** **t**	0.601 ***	0.650 ***

## Data Availability

Not applicable.

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
