# Peer review of "Interferon Tau (IFNt) and Interferon-Stimulated Genes (ISGs) Expression in Peripheral Blood Leukocytes and Correlation with Circulating Pregnancy-Associated Glycoproteins (PAGs) during Peri-Implantation and Early Pregnancy in Buffalo Cows"

_animals, 2022, doi:10.3390/ani12223068_

Round 1
Reviewer 1 Report
I found the article interesting, valuable and clearly written. The methods, statistical analysis and interpretation of the results are correct. I am not a native speaker but I would recommend that the authors do a minor linguistic correction or recheck. In detail:
1. Introduction p.2, l.75 showing change to showed
2. Introduction p.3, l.99 there are not studies should be changed to there are no studies or there are not any studies
3. Materials and Methods p5, l.160 pn the plasma ???
4. Conclusions p12, l.364 further research required change to further research will be required.
Reviewer 2 Report
The manuscript authored by Anna Beatrice Casano et al. describes the Interferon Tau (IFNt) and Interferon-Stimulated Genes (ISGs) expression in peripheral blood leukocytes and correlation with circulating pregnancy-associated glycoproteins (PAGs) during peri-implantation and early pregnancy in buffalo cows.
The submitted paper must undergo a thorough extensive language review.
Specific comments:
Section 1. Introduction:
Please clarify the terminology used.
In line 48, the authors refer to "blastocyst implantation". However, in lines 49 and 50 and for the same period in gestation, "During the peri-implantation, period, the developing conceptus". Are we to understand that blastocyst and conceptus are synonyms for the same stage of embryonic development?
Section 2. Materials and Methods
2.1. Animals and Experimental Design
Please explain better the methodology used.
On line 127, the processing method for the separation of PBMCs and PMNs should be clarified.
Section 4. Discussion
Please explain the differences/similarities of the results concerning previous authors.
In line 311 and subsequent lines, the authors report that the increase in ISG15 expression in PBMCs begins on day 28 post-AI. They also report that these results contrast with those reported by Thakur et al. As mentioned for other parameters, the authors should clarify the reason(s) for this difference. Have the authors used any variation in the methodology employed (e.g. primers) to justify differences in these results?
Reviewer 3 Report
This study is an attempt to explain the “Interferon tau (IFNt) and Interferon-Stimulated Genes (ISGs) expression in peripheral blood leukocytes and correlation with circulating pregnancy-associated glycoproteins (PAGs) during peri-implantation and early pregnancy in buffalo cows”
I listed my other concerns below in the order I found them in the manuscript.
Abstract
The abstract is well written. Please define when the peri-implantation period occurred. It seems that all the associations were done after the implantation. It is clear that INFt and PAGs are produced during early pregnancy otherwise are not produced at all. Why comparing between pregnant and non-pregnant females?
- L27 – L28. When the peri-implantation period occurred?
Introduction
The Introduction is well written. It is clear that INFt and PAGs are produced during early pregnancy otherwise are not produced at all. Why comparing between pregnant and non-pregnant females?
- L92 – L93. PAGs are produced by the trophoblast cells from the blastocyst. Therefore, detecting PAGS 25 d after conception may be an assay detection issue.
- L96 - L97. Why breeding during the low breeding season? Unless the authors provide more information about the reproductive seasonality of the female buffalo, this sentence is out of context.
- L97 – L98. Is there any evidence that the secretion of INFt and PAGs is manipulated by the season? Photoperiod?
- L98. I do not see the relevance of this sentence.
- L99 – L104. Why compare pregnant vs non-pregnant females?
Material and Methods
- L143. Please provide the range reported by other authors.
- L118 – L125. Please provide the month when the synchronization occurred. Was it during the optimal or sub-optimal breeding season? If it was during the sub-optimal breeding season, please justify the season's selection.
Results
- L223 – L228. I do not see the relevance of providing these results.
- L235 – L249. As before, I do not see the relevance of comparing pregnant vs non-pregnant females.
Figures
- Figure 1. I do not see the relevance of this figure.
Round 2
Reviewer 2 Report
Regarding to section 1: introduction, the authors indicate: we have used the term conceptus because this term includes all structures that develop from the zygote, both embryonic and extraembryonic, so we have used "developing conceptus with the meaning of a growing unit.
I personally insist that the term is misused, considering that "blastocyst" also refers to a developing unit. The blastocyst includes embryonic (inner cell mass) and extraembryonic (trophoblast) structures. Therefore, the terminology used should be changed.
Regarding section 2: Materials and Methods
2.1. Animals and Experimental Design
If the methodology is mentioned in line 127 it means that at this point it should be explained in detail, for a correct reading and interpretation of the work. Please make the suggested changes.
Regarding Section 4. Discussion
The changes made adequately clarify the comments.
